# Evaluation of Lake Sediment Thickness from Water-Borne Electrical Resistivity Tomography Data

**DOI:** 10.3390/s21238053

**Published:** 2021-12-02

**Authors:** Johannes Hoppenbrock, Matthias Bücker, Jakob Gallistl, Adrián Flores Orozco, Carlos Pita de la Paz, César Emilio García García, José Alberto Razo Pérez, Johannes Buckel, Liseth Pérez

**Affiliations:** 1Institute of Geophysics and Extraterrestrial Physics, TU Braunschweig, 38106 Braunschweig, Germany; m.buecker@tu-braunschweig.de (M.B.); j.buckel@tu-braunschweig.de (J.B.); 2Research Unit Geophysics, Department of Geodesy and Geoinformation, TU Wien, 1040 Vienna, Austria; jakob.gallistl@tuwien.ac.at (J.G.); adrian.flores-orozco@geo.tuwien.ac.at (A.F.O.); 3Zentralanstalt für Meteorologie und Geodynamik, Section Applied-Geophysics, 1190 Vienna, Austria; 4Geotem Ingeniería S.A. de C.V., Mexico City 14640, Mexico; cpita@geotem.com.mx (C.P.d.l.P.); egg.geotem@gmail.com (C.E.G.G.); arp.geotem@gmail.com (J.A.R.P.); 5Institute of Geosystems and Bioindication, TU Braunschweig, 38106 Braunschweig, Germany; l.perez@tu-braunschweig.de

**Keywords:** geophysics, sediment thickness, electrical-resistivity tomography, water borne, inversion, reflection seismic, sub-bottom profiler, Karst, tropics, lake Nahá

## Abstract

Lakes are integrators of past climate and ecological change. This information is stored in the sediment record at the lake bottom, and to make it available for paleoclimate research, potential target sites with undisturbed and continuous sediment sequences need to be identified. Different geophysical methods are suitable to identify, explore, and characterize sediment layers prior to sediment core recovery. Due to the high resolution, reflection seismic methods have become standard for this purpose. However, seismic measurements cannot always provide a comprehensive image of lake-bottom sediments, e.g., due to lacking seismic contrasts between geological units or high attenuation of seismic waves. Here, we developed and tested a complementary method based on water-borne electrical-resistivity tomography (ERT) measurements. Our setup consisted of 13 floating electrodes (at 5 m spacing) used to collect ERT data with a dipole–dipole configuration. We used a 1D inversion to adjust a layered-earth model, which facilitates the implementation of constraints on water depth, water resistivity, and sediment resistivity as a priori information. The first two parameters were readily obtained from the echo-sounder and conductivity-probe measurements. The resistivity of sediment samples can also be determined in the laboratory. We applied this approach to process ERT data collected on a lake in southern Mexico. The direct comparison of ERT data with reflection seismic data collected with a sub-bottom profiler (SBP) showed that we can significantly improve the sediment-thickness estimates compared to unconstrained 2D inversions. Down to water depths of 20 m, our sediment thickness estimates were close to the sediment thickness derived from collocated SBP seismograms. Our approach represents an implementation of ERT measurements on lakes and complements the standard lake-bottom exploration by reflection seismic methods.

## 1. Introduction

Lakes are in constant contact with their surrounding environment over long periods of time by hydrological, geomorphological, and climatical processes. Therefore, they can be used to reconstruct past environmental and ecological changes [1]. These changes can be reconstructed from fossil microcrustaceans, diatoms, microbes, pollen, and other biological and non-biological components. Deposited and stored on the lake bed building layers of sediment [2,3], these indicators provide high-resolution information on past environmental changes [4,5,6,7,8,9,10]. There is special interest in recovering unaltered and old sediments from ancient lakes to provide a continuous and long historical record of climate and environmental change [11,12,13,14,15]. Geophysical methods help to determine sediment thicknesses prior to coring and drilling campaigns and resolve the inner structure of sedimentary layers (e.g., [16,17]).

Because water, sediment, and bedrock have contrasting mechanical properties, acoustic waves propagate at different velocities within these materials and are reflected at the respective boundaries. Therefore, and due to the straightforward processing and interpretation, reflection-seismic methods are often used to determine sediment thicknesses and generate information of the internal structure of sedimentary sequences at the lake floor (e.g., [18,19]). However, if the seismic velocities (or acoustic impedances) of adjacent layers are too similar or seismic data are noisy due to natural or cultural noise, it might be necessary to apply complementary geophysical methods to provide a more comprehensive view of the lake-floor geology (e.g., [16,20]).

Geoelectrical and electromagnetic methods can also be adapted for water-borne surveys and provide important complementary information (e.g., [17]). These methods are based on the different electrical resistivities of water, sediment, and bedrock. Here, we focused on electrical-resistivity tomography (ERT) (e.g., [21]). Water-borne ERT measurements (with floating electrodes) have already been used to characterize a submarine groundwater discharge [22], a fault zone beneath a river [23], groundwater flow [24], shallow sedimentary sequences in rivers [20,25], lake sediments [26], and river–groundwater interactions [27]. ERT measurements with underwater electrodes have been used to examine rock quality prior to a tunnel-construction project [28].

The basic water-borne setup of the ERT method consists of four floating electrodes for measuring laid out along a profile at the water surface. All electrodes are in direct galvanic contact with the water. Two of the electrodes are used to feed current into the water column, and the other two electrodes measure the resulting voltage. The measured voltage-to-current ratio, or transfer resistance, contains information on the resistivity of the material located below the electrode setup. By varying the electrode positions along the line, in particular by varying the total length of the four-electrode array (i.e., increasing the separation between the current and potential dipoles), the measurement becomes sensitive to the resistivity at different depths. To determine the actual variation in the subsurface resistivity from the measured transfer resistances, inversion algorithms are used to adjust a resistivity model of the subsurface, which explains the measured data (e.g., [21]). ERT data measured along a single survey line are usually inverted to 2D sections of electrical resistivity (e.g., [22,24,26]), i.e., the model consists of rectangular or triangular model blocks, each associated with a resistivity value, which is adjusted during the inversion process. An inherent problem of geophysical inversion is the non-uniqueness of the obtained model: An infinite number of resistivity models can be adjusted to the measured data [29]. If a priori information on the subsurface is available, it may thus be desirable to limit the resistivity of parts of the model to a predefined range or include geometrical constraints, such as known layer thicknesses of boundaries, in order to reduce the ambiguity of the model. A priori information can come, e.g., from borehole data [30] or ground-penetrating radar data [31].

In the case of water-borne measurements, water resistivity (or its inverse, electrical conductivity) can be measured easily with a conductivity meter and water depth can be obtained from simple sonar measurements (e.g., [22]). Most standard 2D-inversion algorithms therefore allow including this information in terms of constraints on conductivity and/or depth of the water layer (e.g., [22,32]), such that only the sub-bottom resistivity distribution is adjusted during the inversion process.

In simple terms, the geology of the lake floor consists of two layers: a sediment layer and the underlying bedrock. Thus, constraining the sediment resistivity during the inversion (in addition to the water resistivity and the depth) may further improve the sediment-thickness estimate. The implementation of a sediment layer of known resistivity on the one hand and a variable thickness on the other hand are not straightforward in standard 2D-inversion software. This problem may be overcome by inverting a series of layered 1D models to the ERT data.

In this study, we adapted the open-source 1D-inversion algorithm by Ekinci and Demirci (2008) [33] to invert water-borne ERT data collected on the tropical lake Nahá in the Karst mountains of the Selva Lacandona, southern Mexico. The main objective of this study was to develop and test a procedure to improve the estimate of the sediment thickness at the lake bottom from ERT data by a constrained inversion. The challenge to resolve a thin (∼1–2 m) conductive sediment layer at relatively large water depths (∼10–20 m) was overcome by constraining the water depth and the conductivity as well as the sediment resistivity in the 1D inversion. To determine the sediment resistivity, we carried out laboratory measurements on sediment samples. The proposed approach was evaluated by comparing the obtained sediment-thickness values with collocated sub-bottom profiler (SBP) data.

## 2. Materials and Methods

### 2.1. Study Site and Survey Layout

The Selva Lacandona rain forest (16–17.5° N; 90.5–92° W; 500–1500 m.a.s.l.) is located in the northeastern part of the Mexican state of Chiapas (Figure 1a). It is part of the Chiapas fold-and-thrust belt generated during the Middle Miocene synchronous to a collision of the Tehuantepec Transform/Ridge with the Middle America Trench off Chiapas [34]. The geology of the Selva Lacandona is characterized by tectonically fractured limestone, which, in combination with the humid subtropical climate, favors Karstification [35]. Limestone caves resulting from calcite solutions by infiltrating rainwater collapse and generate Karst depressions such as dolines, which eventually fill with water and form lakes [17,36].

This study focused on Lake Nahá (Figure 1b) (area ∼0.61 km^2^, max. depth: 32 m, ∼830 m.a.s.l.), which is part of the hydrologically and ecologically important lake system of the Selva Lacandona (e.g., [37,38,39]). During a field campaign in March 2018, we collected reflection seismic data with a SBP and ERT data with a floating electrode setup. In addition, conductivity-probe measurements were carried out at various depths to assess the vertical conductivity gradient across the water column. The water conductivity at a depth of 1.0 m was determined to 460.9 μS/cm and the conductivity at 23.2 m depth to 422.0 μS/cm (Pérez pers. Comm.). The survey covered the entire area of Lake Nahá, whereas only data corresponding to the three eastern sections A, B, and C (Figure 1b) are discussed in more detail in this study. Intersections between ERT and SBP profiles and constant water depths around 20 m make sections A and B well suited for a comparison between the two methods. Section C with a water depth of about 10 m is representative for the shallow areas of the lake and was therefore selected for a more detailed analysis.

The basic approach of our study is illustrated in Figure 2. The focus was on the highly constrained inversion of ERT data, which uses a priori information either collected simultaneously with the ERT measurements on the lake (water resistivity and water depth) or subsequently in the laboratory (sediment resistivity). These ERT-derived sediment-thickness estimates were then compared to the sediment thickness obtained from SBP profiles at the same locations.

### 2.2. Sub-Bottom Profiler (SBP)

#### SBP Data Acquisition

The StrataBox HD by SyQwest is a low-frequency (10 kHz) echo sounder, also referred to as a sub-bottom profiler (SBP), which can be used to image sedimentary layers down to a depth of some tens of meters below the lake bottom (e.g., [18,40]). The measurement system emits a pulsed seismic wave (or sound wave) into the water and records the reflected waves traveling back to the surface (Figure 3a). Because of the contrasting seismic velocities of water, sediment, and limestone bedrock, seismic waves are (partly) reflected at the limits between these units [18]. A single SBP measurement consists of the recorded amplitude of the wave over time, also known as a seismic trace (Figure 3b). As the travel time between the surface and a specific reflector only depends on the wave velocity and the distance between the sensor and the reflector, it is possible to reconstruct depths and thicknesses of different layers. A SPB profile consists of many seismic traces acquired at closely spaced points along a profile crossing the lake. These data are usually visualized in terms of a seismogram (Figure 3c), which gives a good first impression of the geometrical configuration of the sediment layers at the lake bottom.

In this study, the sensor of the measurement system was mounted mid-ship in a side-mount configuration at 0.4 m below the water surface. SBP data were collected on a regular grid of NS- and EW-oriented lines (see Figure 1b). The exact trajectory of boat and sensor were tracked with a differential GNSS mounted on the boat, and GNSS data were stored along with the SBP data. SBP acoustic traces were visualized using the free GSEGYView (version 0.2) software provided by Bashkardin (2013) [41]. As the SBP raw data had a high overall quality, no further processing steps had to be applied (e.g., to deal with noisy traces).

### 2.3. Electrical-Resistivity Tomography (ERT)

#### 2.3.1. ERT Data Acquisition

The water-borne electrical-resistivity tomography (ERT) measuring system consisted of a 65 m-long marine resistivity cable with 13 stainless-steel electrodes, which was dragged by a motorboat (Figure 4). A Syscal Pro Switch-48 device (Iris Instruments) was used for continuous data acquisition while the boat was navigating along the measuring lines. Water depth (at the position of the boat) and boat position were continuously measured by a simple echo sounder with an integrated GNSS and stored along with the ERT data.

The first electrode of the cable was located 5 m behind the boat, and the unit spacing between the electrodes was 5 m. The cable was kept afloat by 20-cm long pieces of insulation-foam pipes, positioned between the electrodes. We used a dipole–dipole (DD) configuration, with the current being injected into the water by the last two electrodes, i.e., between 60 and 65 m from the boat, and the potential differences being measured between every other pair of adjacent electrodes. Using a DD configuration takes advantage of the full number of ten channels of the measuring device and speeds up the sounding data acquisition compared to configurations like Wenner or Schlumberger, which only use one channel at a time (e.g., [42]). This is particularly important for water-borne surveys, during which the electrode cable is moving continuously. We collected both electrical-resistivity and time-domain-induced polarization data (not discussed in this study) with a maximum injected current of 40 mA. All measurements were carried out with a constant number of three stacks. At each location of the current dipole, the measured data consisted of the injected current (*I*) and ten measured voltage values (*U*), which correspond to different investigation depths. Figure 5 shows a sketch of a typical pseudo-section showing the data points corresponding to measurements at various current dipole locations. The geometry factor for the dipole–dipole array is (e.g., [43]).
(1)k=πn(n+1)(n+2)a
where *a* is the unit electrode spacing, and *n* is the distance between current and voltage dipole in terms of unit spacings. The geometry factor allows converting the measured voltage-to-current ratio into an apparent resistivity (e.g., [43]).
(2)$$\rho_{a}=k*\frac{U}{I}.$$

A complete ERT profile consists of measurements carried out at many different locations of the current dipole. The results can be visualized in terms of a pseudo-section (Figure 5). During data acquisition, the boat was navigating along the survey line at velocities between 0.5 and 1 m/s. As the average duration of one measurement was 20 s, the distance between two adjacent measurements varied between 10–20 m.

#### 2.3.2. Processing and Inversion of ERT Data

We used two approaches to obtain the input sounding data (i.e., apparent resistivity vs. dipole distance) from our continuous ERT measurements along the survey lines:The 10 apparent resistivity measurements at one single current–dipole location.The mean values of the 10 apparent resistivity measurements of *x* consecutive current–dipole locations (as sketched in Figure 5 for x=4).

The first method is preferable, as the averaging of the apparent resistivity of various measurements along the line reduces the lateral resolution. However, averaging several measurements may also significantly improve the signal-to-noise ratio. This becomes more important the deeper the water is. Moreover, if both water depth and sediment thickness do not vary considerably over a longer distance, the use of the second method is justified. The decision on how many data points are averaged at a specific location depends on the local data density and quality. For the purpose of this study, this decision was taken after a manual inspection of the data.

The low signal-to-noise ratio at larger dipole distances poses restrictions on the interpretation of ERT data from deep-water zones. Figure 6 shows the sounding data of three adjacent current–dipole locations of section A (see Figure 1b) at a water depth of ∼20 m. Although the bathymetry is flat in this area, the measured apparent resistivity scattered significantly, particularly at larger dipole distances. Before we computed averages of the apparent-resistivity values for each dipole distance separately to obtain our sounding curve, in this study we manually eliminated obvious outliers (see Figure 6).

After filtering and averaging, the sounding curves were inverted using a modified version of the damped least-squares inversion algorithm VES1dinv described by Ekinci and Demirci (2008) [33]. This free algorithm, which is implemented in MATLAB^®^, minimizes the squared misfit between the calculated and the true sounding curve with the help of a singular-value decomposition of the Jacobian matrix. It stops when the misfit no longer decreases, has reached a minimum value, or the maximum number of iterations has been reached. Originally, the algorithm was created for the forward computation (and inversion) of the apparent resistivity for the Schlumberger configuration. Because the forward computation is based on a digital filter for the Hankel transform, the adaptation of the algorithm to our needs was readily accomplished by using suitable filter coefficients for the dipole–dipole configuration [44]. Furthermore, the possibility to set constraints was added in the algorithm by setting the corresponding entries in the Jacobian matrix to zero. The modified algorithm is published with this manuscript [45]. To evaluate the expected improvement of the sediment-thickness estimate obtained from the constrained 1D inversion, we also carried out a smoothness-constrained least-squares 2D inversion using RES2DINV [32]. This software allows constraining water depth and water resistivity for water-borne surveys. In order to allow for a sharp resistivity contrast between water and sediment, the smoothness constraint at the corresponding interface was relaxed.

The quality of fit of both 1D and 2D inversions is usually expressed in terms of the root-mean square (RMS) error, which we computed as the mean-squared deviation between the synthetic and the measured apparent resistivity values.

#### 2.3.3. Forward Modeling Study

To further assess the theoretical potential of our ERT measurements for the detection of sediment layers at different water depths, we carried out a modeling study. Forward models for different water depths and thicknesses of a conductive sediment layer (introduced in more detail in the results and discussion section) were computed using the free software RES2DMOD [46]. Synthetic apparent-resistivity data were generated for a dipole–dipole configuration with 13 electrodes at a 5 m spacing. Apparent-resistivity values were obtained for complete pseudo-sections, i.e., all possible consecutive current–dipole positions along the line were modeled. To account for the increase in the noise level with the distance between the current and the potential dipole, synthetic normally distributed noise was added to the apparent-resistivity data. We used 0% noise for array sizes n∗a≤25 m, 3% for array sizes 25m<n∗a≤40 m, and 5% for array sizes n∗a>40 m. To assess the detectability of the sediment layer, 2D and 1D inversions were conducted with different constraints on the inversion.

### 2.4. Laboratory Measurement of Sediment Resistivity

With the aim of implementing sediment resistivity as an additional constraint during the inversion, we determined the electrical-resistivity of sediment samples in the laboratory. For these measurements, we used the 49 cm-long core Nah-16 I, which was recovered in June 2016, roughly 2 years before the water-borne survey. The approximate core-recovery location of Nah-16 I (16°59′8.33″ N, 91°35′31.82″ W, see Figure 1 ) is located in the deepest part of Lake Nahá and was chosen based on bathymetric data. Subsequently, the short sediment core was sampled every 0.5 cm (Figure 7a), packed into small sealed plastic bags (Figure 7b), and kept cool during transport and storage.

To measure its resistivity, the core material was filled into a four-point measuring cell (Figure 7c) with a diameter of 2.5 cm and a length of 4.5 cm (see [47]; for details on the measuring cell). To fill the measuring cell, we used an average of 14 subsamples, which corresponds to ∼7 cm of the core. Samples from three different sections of the core (0–7 cm, 7–14 cm, and 40–49 cm) were analyzed. We used material from the top and the bottom of the core because we expected the largest variation in resistivity between these two sections due to different compositions. In order to avoid errors related to different packing strategies of the altered sample material, the sediment was installed in three different ways: (i) uncompressed, (ii) compressed, and (iii) saturated with water that had a conductivity of 26 Ωm. As it is impossible to recreate the real conditions of the unaltered sediments at the lake bottom, this approach was meant to narrow down the range of sediment resistivity values as far as possible. The measurements were carried out with a Chameleon I (Radic Research) electrical-impedance analyzer [48] at different frequencies of the injected current between 1 mHz and 1 kHz. At least 24 h before the actual measurement, the filled sample holder was stored in a climate chamber at 20 °C in order to keep the sample at a well-defined and constant temperature during the measurement.

## 3. Results and Discussion

### 3.1. SBP Data Reveals Bathymetric Profile and Sediment Thickness

The bathymetry (lakebed topography) along the three north–south (and south–north) running SPB profiles displayed in Figure 8 showed typical features of the Karst lakes of the region (e.g., [17]): A mostly flat bottom with steep walls and cliffs along the shore line or between sections of the lake with equal water depth. For these three profiles, the maximum water depth of 21.1 m was observed in the central part of L3 NS (Figure 8b). The maximum water depth of ∼32.0 m for the entire water body was detected close to the intersection of SBP lines L2 SN and L6 WE in the southern part of Lake Nahá (see Figure 1; seismogram not shown for brevity).

Within the steep parts of the profiles, the reflection seismograms only showed one strong reflection. According to Bücker et al. (2021) [17], who discussed SBP data from nearby lakes Tzibaná and Metzabok, these parts correspond to uncovered limestone-bedrock outcrops at the lake bottom. Across the flat parts of the profiles, we observed one or various parallel reflector horizons indicating the presence of sediment layers covering the limestone bedrock. Along the three SBP profiles, the thickness of these sediment layers ranged between ∼1.5 m in the shallow (∼10 m) and ∼2.5 m in the deeper (∼20 m) areas of the lake.

### 3.2. Laboratory Measurements Narrow down Sediment Resistivity

The resistivity of the sediment samples of core Nah-16 I show that the resistivity values varied between 8.5 and 11.0 Ωm within the studied frequency range (Figure 9). The sample resistivity varied with both the packing procedure and the core depth. For the 0–7 cm sample, which was packed in three different ways, the resistivity varied over the entire range from ∼8.5 Ωm for the uncompressed to 11.0 Ωm for the compressed sample. If packed with the same procedure (saturated with water), the sample resistivity increased from ∼9.0 Ωm for the 0–7 cm sample to ∼10.0 Ωm for the other two samples. Within the sediment layer at the lake bottom, we expected different degrees of compaction (as introduced by different packing procedures) and compositional variations (as observed in the different sections of the studied core). In the following, we therefore use a relatively broad range of 8.0–12.0 Ωm as the upper and lower limit for the inversion of our field data. The results are consistent with the study by Bücker et al. (2021) [17], who reported sediment-resistivity values between 10 and 15 Ωm for fine sediments recovered from nearby lakes.

### 3.3. Modeling Study Shows That Constrained 1D Inversion Improves Sediment-Thickness Estimate from ERT Data

Based on the bathymetric information and sediment-thickness estimates obtained from the SBP profiles (Figure 8), two synthetic resistivity models were used for the modeling study: (i) a sediment layer of 1.5 m and a water depth of 10.5 m (Figure 10a) and (ii) a sediment layer of 2.5 m and 21.0 m water depth (Figure 10b). Water resistivity was set to 26.0 Ωm, which roughly corresponds to the average conductivity of Lake Nahá reported by Rubio (2016) [36]. Based on the average resistivity of our laboratory measurements, sediment resistivity was set to 10.0 Ωm. For the limestone bedrock, we assumed a resistivity of 200.0 Ωm (e.g., [49]).

From this synthetic model, a forward model was computed as described in Section 2.3.3. Figure 11 shows the results of a 2D inversion with RES2DINV for both synthetic models. The depth and the resistivity of the water layer were fixed to the known values of the respective original models. In the inversion of the model with a water depth of 10 m (Figure 11), both the limestone bedrock and the sediment layer appeared as separate layers. However, the thickness of the sediment layer was largely overestimated compared to the layer thickness of the original model. This is an inherent weakness of the method known as the principle of equivalence: Only the quotient $\frac{t}{\rho }$, where *t* is the layer thickness and ρ is its resistivity, can be determined for a thin, conducting layer [50]. In the inverted model for a water depth of 21 m (Figure 11b), resistivity values only decrease weakly below the water layer, and the limestone bedrock was not detected at all.

In order to improve the sediment thickness estimate for both water depths, we used the 1D inversion code VES1DInv, which allows for constraining the resistivity of the sediment layer in addition to the constraints on water depth and water resistivity. Figure 12 shows the inversion results for the model with 21 m water depth for a sediment resistivity of 8.0 and 12.0 Ωm (min. and max. values of the sediment resistivity measured in the laboratory), which resulted in a sediment thickness of 1.61 and 2.24 m, respectively. The original model represented by the black line had a sediment resistivity of 10.0 Ωm and a thickness of 2.5 m. These observations indicate that fixing the sediment resistivity to a narrow range may clearly improve the sediment thickness estimate—even in the case of 21 m of water column. It is obvious that the same procedure works even better for the model with the 10.5 m water column (results not shown for brevity). Another observation was that the resistivity of the limestone layer below the sediment was not correctly resolved, and, in this case, the value was overestimated. The resistivity of the limestone layer changes largely depending on the start-model resistivity and the noise level.

Our modeling study indicates that the highly constrained inversion approach allows estimating the sediment thickness even under conditions where the water depth is larger than 1/3 of the maximum dipole separation.

### 3.4. Constrained 1D Inversion Yields Sediment-Thickness Estimates Comparable to SBP Results

Based on the promising results of the modeling study, in this section, we apply our constrained 1D inversion to measured data. To validate the inversion results with collocated SBP data, we focused on ERT data collected at intersections with SBP survey lines, where the water depth did not significantly vary along the ERT lines (see Figure 8). At these locations, it was possible to compare the ERT-derived sediment thickness with the layering in the SBP seismogram, and the use of a 1D inversion approach was well justified.

#### 3.4.1. Performance in Deep Water

As an exemplary dataset for a deep-water situation, we selected the ten dipole positions of section B along ERT line Nah9 (see Figure 1b). From SBP profile L4 SN (Figure 8a), which is parallel to ERT line Nah9, we observed that the water depth was nearly constant in this section. For each dipole distance n·a, the apparent resistivity values in Figure 13a were computed as the corresponding mean values of all ten current dipole positions of section B (see Figure 1b). In the following, the water resistivity was determined by an inversion of the first four potential values (i.e., n≤4), which were mainly influenced by the water layer. In this section, the water resistivity was fixed to 22.8 Ωm. The sediment resistivity was fixed to either 8.0 Ωm or 12.0 Ωm, which are the minimum and maximum resistivity determined in the laboratory, respectively (Figure 13).

Following the same procedure as in the modeling study, the inversion yields two resistivity models with a sediment thickness, which depends on the fixed resistivity of the sediment layer (Figure 13b): 2.21 m for a resistivity of 8.0 Ωm and 3.61 m for a resistivity of 12.0 Ωm, respectively. From the collocated SBP profile, we obtained a sediment thickness of 2.5 m (see Figure 8a) at this position, which falls within the range of possible sediment thickness estimates from the inversion of the ERT data.

Although the total RMS error of 2.86% of the inversion was relatively small, the variance in the apparent resistivity values in the dataset of ten current–dipole locations (shown in terms of error bars in Figure 13a) increased significantly with the dipole distance. However, the comparison of the inversion results with the collocated SBP data in Figure 13b indicates that the sediment thickness can be determined within a realistic range.

To determine the influence of possible inaccuracies of the constraints included in the 1D inversion (i.e., water depth and water resistivity), we carried out an additional parameter study. For this purpose, the constraints on water depth and on water resistivity were varied, while all other constraints were kept constant. The maximum error of the water-depth measurement (and thus the range of the variation of this constraint in the parameter study) was estimated to be 1 m, and the maximum error of the water resistivity was approximated by 0.5 Ωm. A resistivity of 10.0 Ωm was assumed for the sediment layer. In a further step, models were generated in which all constraints—water depth and resistivity and sediment resistivity—were modified to assess the maximum expected deviation of the sediment thickness caused by a combination of both errors.

Figure 14 shows that using an erroneous water resistivity as a constraint had the strongest influence on the sediment thickness. While the sediment thickness was underestimated if the water resistivity was assumed to be lower than it actually was, it was overestimated for an increased water resistivity (Figure 14a). In contrast, an erroneous water depth only changes the vertical position (i.e., the depth under the water surface) of the layer but not significantly the sediment thickness (Figure 14b). A combination of erroneous constraints can greatly distort the resulting sediment layer thickness (Figure 14c). Here, the water resistivity had the greatest influence again.

In Figure 15, we present inversion results for the three groups of current–dipole locations of section A along ERT line Nah13 (see Figure 1) to evaluate the consistency between the models obtained at different locations. Each of the three groups corresponds to an intersection of the west–east-running ERT line with one north-south-running SBP profile. According to the SBP profiles (Figure 8a–c), the water depth and the sediment thickness slightly varied between the intersections. The inversion results in Figure 15 show that the “true” SBP-derived sediment thickness did not differ significantly from the ERT-based values at any of the three intersections. The reduction in the RMS value of the inversion is probably a result of the decreasing water depth (from left to right in Figure 15).

#### 3.4.2. Performance in Shallow Water

ERT data from shallow areas were available for the northern and southern parts of the lake. We selected section C (see Figure 1b) located in the southeastern part of Lake Nahá to assess the applicability of our approach to shallow-water situations. SBP profile L4 SN (Figure 8a), which crosses the corresponding ERT line Nah6, showed a thin sediment layer of 1.5 m in a water depth of 10.5 m. The one-dimensional inversion of the averaged ERT data with the constraints on water resistivity and depth as well as sediment resistivity resulted in the 1D models shown in Figure 16. The sediment thickness obtained for sediment resistivities of 8 and 12 Ωm were 0.84 m and 1.26 m, respectively. At least the second value was close to the value of 1.5 m estimated from the collocated SBP profile.

### 3.5. 2D Inversion of Shallow-Water ERT Data Were Not Able to Estimate Sediment Thickness

Our modeling study shows that a common 2D inversion with a constrained water layer was able to resolve the sediment layer in shallow water. However, as constraining sediment resistivity is not straightforward in RES2DINV, the thickness of the sediment layer obtained in the modeling study was largely overestimated. Although, in general, it is possible to constrain the sediment resistivity in 2D with other inversion packages, e.g., pyGIMLI [51], we decided to compare our approach to RES2DINV, as this is the standard software in the industry by now. In the following, we briefly discuss how the inversion result of the standard 2D inversion compares to our constrained 1D-inversion approach.

Based on the position data of the current dipole, the sounding data of section C were first arranged in terms of a pseudo-section (Figure 17a) and then inverted using RES2DINV. During the inversion, a constant water depth of 10.5 m and a fixed water resistivity of 23 Ωm were used as a priori information. Figure 17 shows the resulting resistivity model. In the central part, the inverted model showed a conductor with resistivity values as low as 15 Ωm, which was approximately 5 m thick. The outer parts of the section were not considered, as the data coverage was low.

As expected, the 2D inversion also overestimated the sediment resistivity in the case of field data (compared to laboratory measurements) and compensated the higher resistivity of the sediment layer by simultaneously increasing its thickness. This resulted in a greatly overestimated sediment thickness of 5.0 m compared to 1.5 m as derived from the SBP profile or 0.84 to 1.26 m as obtained from the constrained 1D inversion.

### 3.6. Assessment of the Approach

In this study, the resistivity of the limestone bedrock below the sediment layer was significantly higher (>50 Ωm) than the resistivity of the sediment layer (8–12 Ωm). Thus, the local geology, and particularly the sufficiently high contrast between the sediment and the bedrock was a necessary condition for the successful application of the electrical method. While this condition was fulfilled in Lake Nahá, the practicability of the proposed method at other locations will depend on the specific resistivitiy values of the water, the sediment, and the bedrock.

In a similar way, the flat bathymetry of Lake Nahá and the gentle variation in the thickness of the sediment layer at its bottom favor the interpretation of ERT data with a 1D-inversion algorithm. Lakes with abrupt bathymetric variations and/or large variations in sediment thickness (e.g., [52]) would probably result in inadequate results of such a simple 1D approach.

There are some other assumptions implicit to our approach, which cannot be neglected and which might be a source of inaccuracies. For instance, we assumed that the water layer can be characterized by a homogeneous resistivity. If the water column is sufficiently well mixed, this may be the case. However, in general, lakes display a stratification with respect to temperature, salinity, and biomass, among others [53,54], which can result in a significant variation in the electrical resistivity across the water column. In principle, this variation can be measured with a conductivity–temperature–depth (CTD) probe (or similar) and readily included into our 1D inversion. In the opposite case, the layered conductivity structure of the water column may result in lower conductivity contrasts of water and sediment and may even affect the sediment thickness estimate.

Furthermore, the trajectory of the boat, and consequently the geometry of the cable towed by the boat, may deviate from a straight line, which leads to variations in the geometric factor. In this study, we focused on relatively straight sections of the ERT lines. However, Figure 1b shows that there are survey lines that are considerably curved—especially along the lines, which follow the shoreline. If applied to ERT data from such sections, neglecting the non-linear array geometry could also lead to erroneous interpretations.

In similar studies, either only sediments in shallow water at depths of up to 10 m were investigated (e.g., [20,22,26]) or sediments in deep water were investigated with underwater electrodes on the lakebed [55]. In these studies, the water depth and the water resistivity were partly constrained (e.g., [20,22]) but not the sediment resistivity. Constraining the sediment resistivity is new and allows the determination of sediment thickness at greater depths than 10 m.

In our study, the use of the 65 m cable limited the maximum dipole distance to 55 m. The interpretation of the results was possible down to a water depth of ∼20 m, if the sediment thickness was of the order of 2 m. Compared to the 2D inversion with only the water layer as a constraint, the use of our highly constrained 1D inversion already resulted in a considerable improvement in the depth of investigation and the ability to solve for reasonable sediment thicknesses. Depending on the water depth of the target lake and the sediment thickness, the maximum depth of investigation could be increased further by increasing the electrode spacing, i.e., the length of the cable, as the system is completely scalable [50]. However, the minimum detectable thickness of the sediment layer is expected to decrease with increasing depth of investigation.

## 4. Conclusions

We presented an approach to determine the total thickness of lake-floor sedimentary deposits from water-borne ERT data. Our approach was based on the use of a Matlab 1D-inversion algorithm for the interpretation of ERT data, which allows fixing the water resistivity and the depth as well as the sediment resistivity. A modeling study illustrated the fundamental applicability of the chosen approach, and a case study on the Karst Lake Nahá in southern Mexico showed its practical feasibility.

In particular, our results indicate that it is possible to correctly estimate thin sediment layers ( 1/10 of water depth) down to 1/3 of the maximum length of the employed dipole–dipole layout. These numbers show that the highly constrained inversion significantly pushes the limits of the ERT method compared to standard (2D) inversion approaches. Obviously, the proposed procedure depends on the availability of independent measurements of water resistivity and depth as well as sediment resistivity. The first two values can be obtained using a standard conductivity probe and an echo sounder; a basic procedure for the measurement of the sediment resistivity with a four-point measuring cell in the laboratory is described in this study.

While the maximum depth in Lake Nahá was 32 m, in this study, we restricted ourselves to areas with depths between 10 and 20 m. In these areas, sediment thicknesses between 1.0 and 2.5 m were determined with our ERT measurements. The sediment resistivity was determined in the laboratory in a range between 8 and 12 Ωm. In comparison, the sediment thicknesses obtained by the SBP were also in the range between 1.5 and 2.5 m.

Future developments of the approach could address the incorporation of conductivity-depth profiles of the water column, which includes information about possible variations of the electrical resistivity over the water column. In order to improve the application of the approach to lakes with stronger bathymetric variations and lateral variations of the sediment thickness, it would be desirable to further develop the approach into a laterally constrained 3D inversion. An enhanced version of the algorithm, which allows to include the curved shape of the cable (and thus the deviation of the inline dipole-dipole configuration), is currently in preparation.

## Figures and Tables

**Figure 1 sensors-21-08053-f001:**
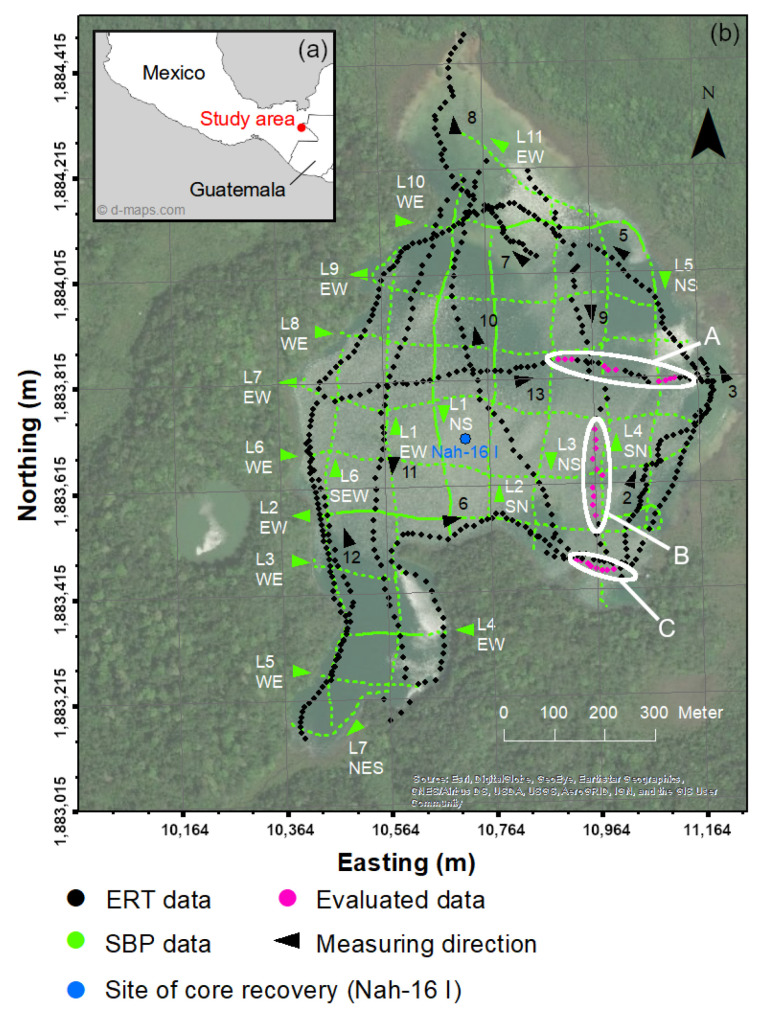
(**a**) Location of the study site in southern Mexico (from d-maps.com, accessed on 26 November 2021). (**b**) Satellite image of Lake Nahá and its surroundings and layout of the water-borne geophysical survey. All electrical-resistivity tomography (ERT) and sub-bottom profiler (SBP) lines of the survey are displayed (small triangles next to the line names indicate the direction, in which the measurements were carried out). Current–dipole locations on the ERT profiles Nah6, Nah9, and Nah13 printed in pink were analyzed and discussed in more detail in this study. The selected current–dipole locations were grouped into three sections, where sections A and B correspond to deep (∼20 m) zones of the lake, and section C represents a shallower (∼10 m) zone. The recovery location of core Nah-16 I, which was used to measure sediment resistivity in the laboratory, is also indicated. (Source: Esri, DigitalGlobe, GeoEye, Earthstar Geographics, CNES/Airbus DS, USDA, USGS, AeroGRID, IGN, and the GIS user community).

**Figure 2 sensors-21-08053-f002:**
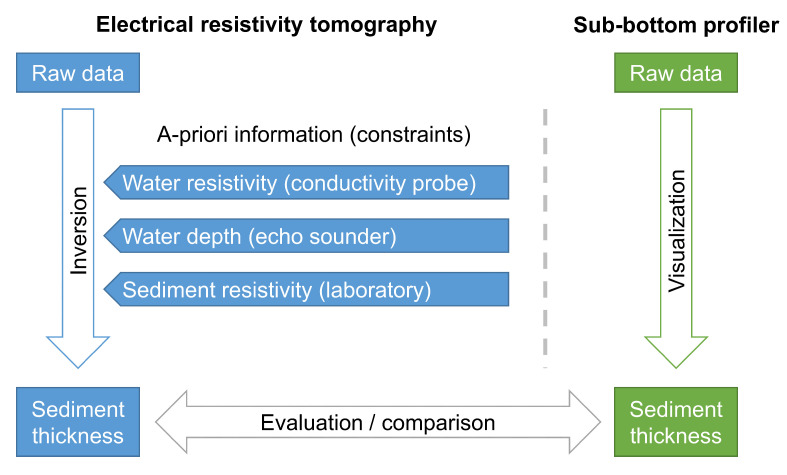
Flowchart describing the general approach of this study. The measured ERT data were inverted by using water depth, water resistivity, and sediment resistivity as constraints. The sediment thickness resulting from the inversion was then compared to the sediment thickness obtained from the SBP profiles.

**Figure 3 sensors-21-08053-f003:**
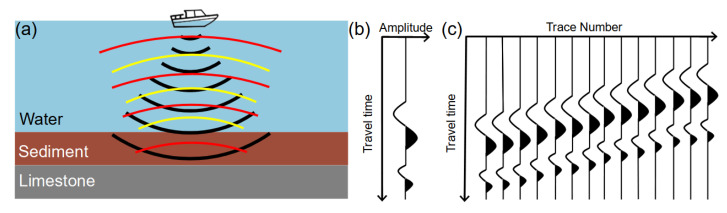
Schematic sketch of the principle of sub-bottom profiling. (**a**) An acoustic pulse is emitted at the water surface and travels downward (black). Waves are reflected at the water-sediment limit (yellow) and the sediment-bedrock interface (red). Both reflected waves travel back to the water surface, where they are recorded at different times. (**b**) In this simple case, the recorded signal shows two peaks corresponding to the two interfaces. (**c**) A reflection seismogram consists of many traces recorded at different positions along a SBP profile.

**Figure 4 sensors-21-08053-f004:**
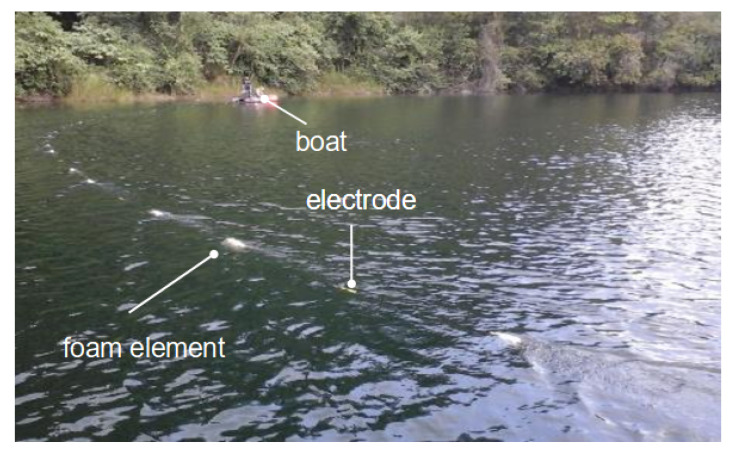
Picture of the water-borne electrical-resistivity tomography (ERT) measuring system. The marine resistivity cable with 13 stainless-steel electrodes in the foreground was kept afloat by foam elements attached to the marine cable. The boat with the measuring device in the background also carried the integrated GNSS and echo-sounder device.

**Figure 5 sensors-21-08053-f005:**
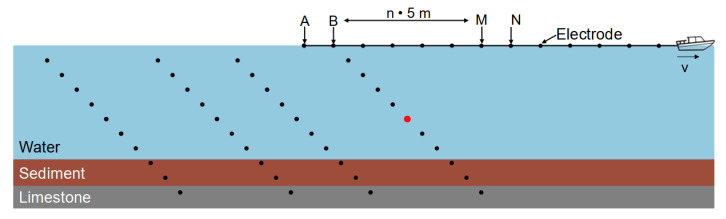
Schematic illustration of the construction of a pseudo-section plot for various subsequent measurements at different injection-dipole positions realized by moving the boat to the right. The boat with the towed cable shows how the pseudo-section is formed with a fixed current dipole position (A, B). The red point marks the position of the measurement taken by electrodes M and N. Note that this is only a schematic illustration; besides the maximum electrode separation on the cable, the actual water depth as well as the electrical properties of the water and the lake-bottom materials largely affect the ability of the system to detect a thin sediment layer.

**Figure 6 sensors-21-08053-f006:**
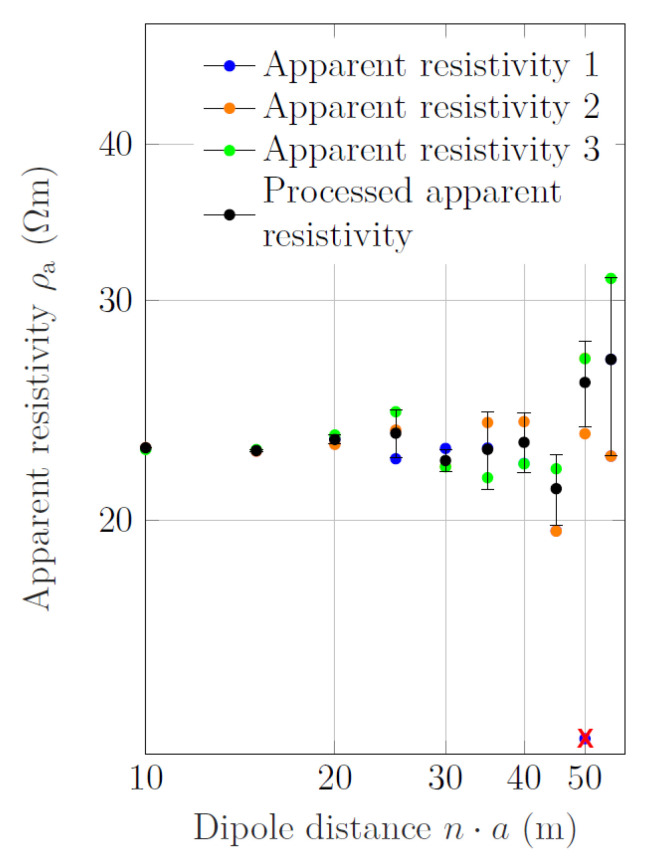
Measured apparent resistivity for three adjacent current–dipole locations in section A (colors) and average sounding curve (black). The error bars indicate the variance of the averaged data points. The water depth in this area was ∼20 m. Before the average apparent resistivity was computed for each dipole distance separately, obvious outliers were removed. In this example, the lowest apparent resistivity of the blue sounding curve 1 at dipole distance 50 m was removed prior to averaging (as indicated by the red x).

**Figure 7 sensors-21-08053-f007:**
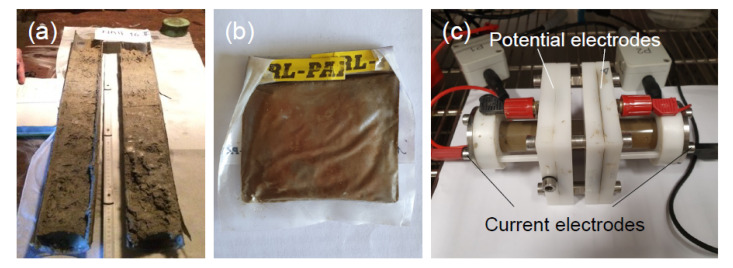
(**a**) Sediment core Nah-16 II, a parallel core of Nah-16 I, directly after recovery (photo: K. Rubio). (**b**) Plastic bag with sediment sample (photo: J. Hoppenbrock). (**c**) Sediment-filled measuring cell (photo: J. Hoppenbrock); current was injected by two stainless-steel caps at the right and the left end of the cell, which were in direct contact with the sample material; the potential difference was measured between two ring electrodes outside of the sediment-filled volume, which were separated from the sediment by a small water reservoir.

**Figure 8 sensors-21-08053-f008:**
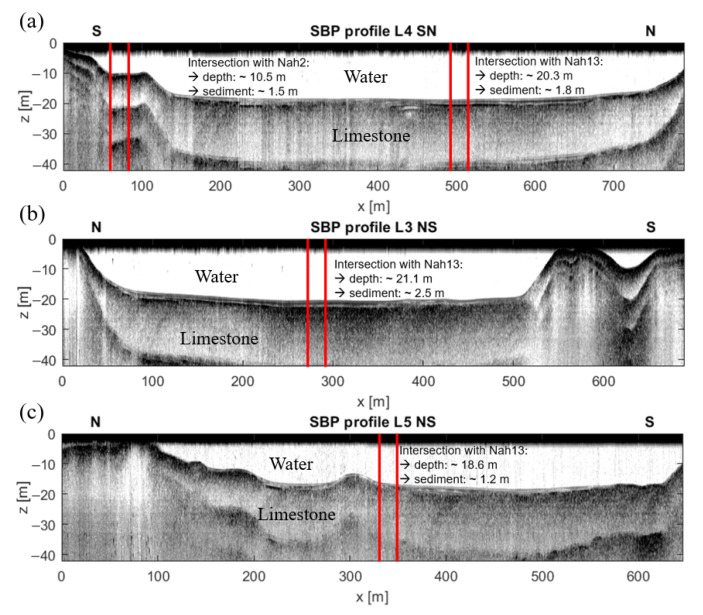
SBP reflection seismograms for three profiles crossing the lake Nahá in direction South–North and North–South, respectively. Red lines mark intersections of SBP and ERT profiles. (**a**) SBP profile L4 NS, which intersects ERT profiles Nah2 and Nah13 and where we observed a sediment layer of 1.5 m at 10.5 m depth (intersection Nah2) and a sediment layer of 1.8 m at 20.3 m depth (intersection with Nah13). (**b**) Along SBP profile L3 NS, the sediment thickness was almost constantly 2.5 m at a depth of 21 m. (**c**) At the intersection with ERT line Nah13, SBP profile L5 SN detected a sediment layer of 1.2 m in 18.6 m water depth.

**Figure 9 sensors-21-08053-f009:**
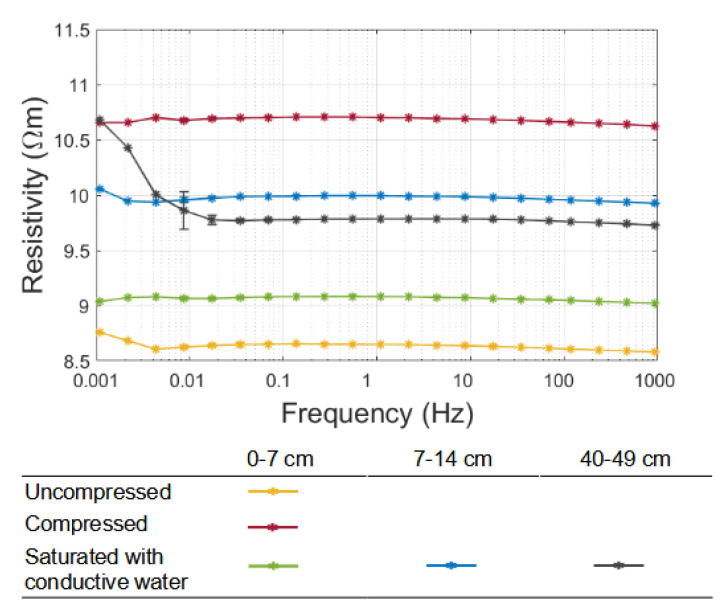
Frequency-dependent resistivity of the sediment samples of core Nah-16 I. Samples from 0–7 cm of the core were packed using different procedures (uncompressed, compressed, and saturated with conductive water). Samples from different sections of the core (0–7 cm, 7–14 cm, and 40–49 cm) were packed using the same procedure (saturated with water).

**Figure 10 sensors-21-08053-f010:**
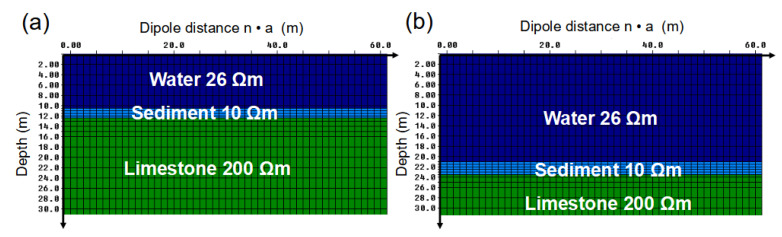
Resistivity models used for the modeling study: (**a**) Model with a 1.5 m sediment layer at 10.5 m water depth. (**b**) Model with a 2.5 m sediment layer at 21.0 m water depth.

**Figure 11 sensors-21-08053-f011:**
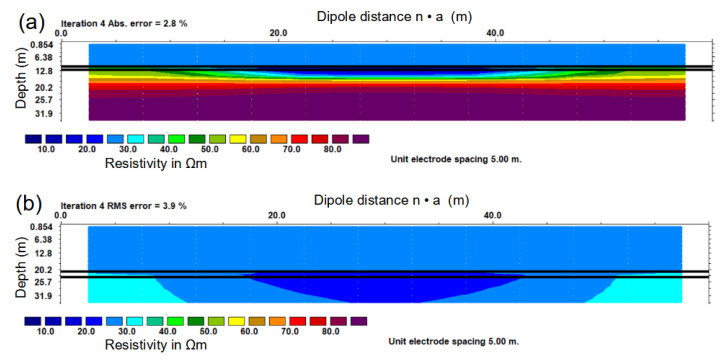
Inversion of synthetic data for (**a**) a sediment layer of 1.5 m and 10.5 m water depth and (**b**) 2.5 m sediment layer and 21.0 m water depth. The black horizontal lines mark the upper and lower limits of the sediment layer of the original resistivity model.

**Figure 12 sensors-21-08053-f012:**
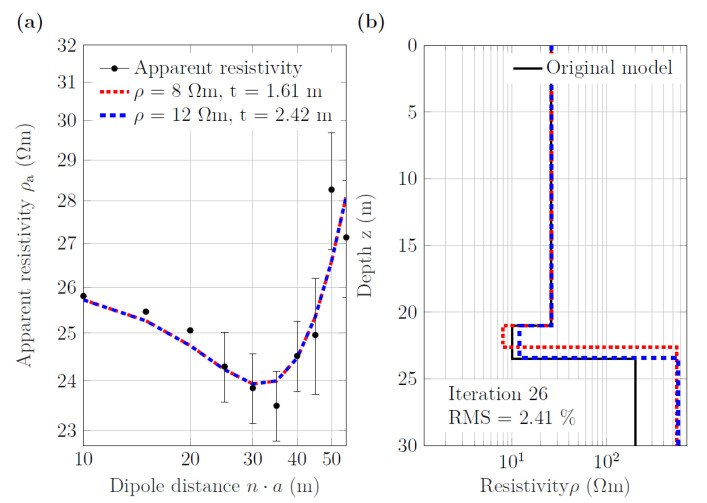
One-dimensional inversion of the synthetic data obtained for the sediment layer at 21.0 m depth including synthetic noise. (**a**) Observed (black dots) and calculated apparent-resistivity sounding curves. (**b**) Layered resistivity models for two different values of the fixed sediment resistivity; RMS errors were the same for both models. The black solid line represents the original model (sediment layer: 10.0 Ωm, 2.5 m).

**Figure 13 sensors-21-08053-f013:**
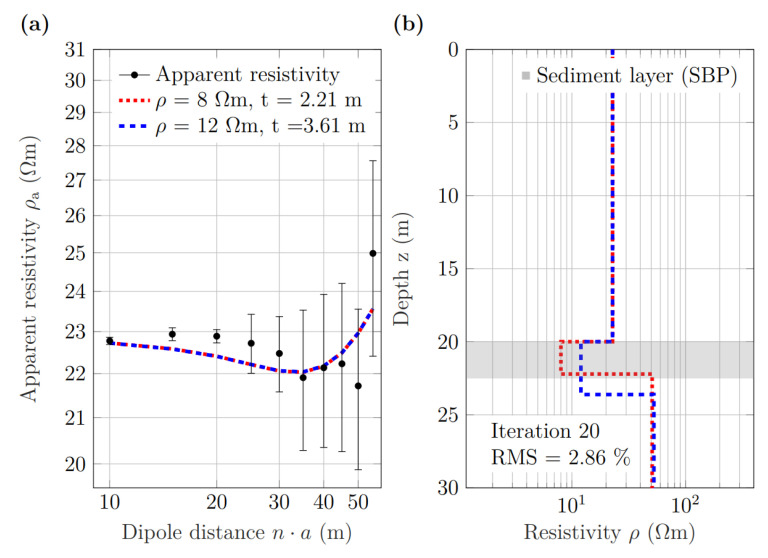
One-dimensional inversion of measured ERT data corresponding to the ten current dipoles of section B along ERT line Nah9. (**a**) Observed (black dots) and calculated apparent-resistivity sounding curves. (**b**) Layered resistivity models for two different values of the fixed sediment resistivity; RMS errors were the same for both models. The grey shading indicates the sediment-layer thickness derived from the collocated SBP line.

**Figure 14 sensors-21-08053-f014:**
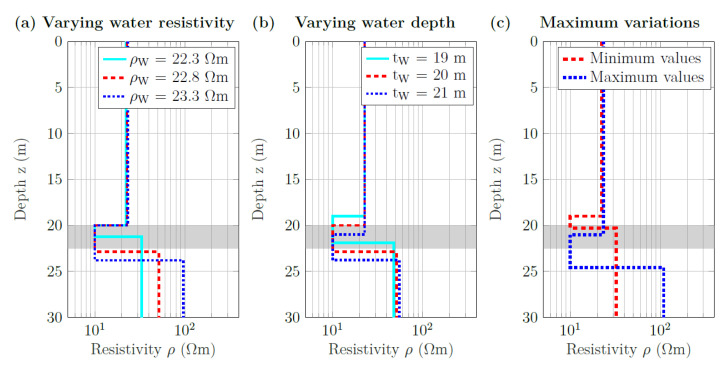
Parameter study to assess the influence of constraints on the inversion of averaged ERT data of section B. (**a**) Water depth ($t_{W}$) and sediment resistivity ($\rho_{S}$) were set to 20 m and 10.0 Ωm, respectively. Water resistivity ($\rho_{W}$) varied between 22.3 Ωm and 23.3 Ωm. (**b**) Water resistivity and sediment resistivity were set to 22.8 Ωm and 10.0 Ωm, respectively. Water depth varied between 19 m and 21 m. (**c**) Layered models with the largest deviations from the true model. For the minimum values: $\rho_{S}$ = 10.0 Ωm, $\rho_{W}$ = 22.3 Ωm, and $t_{W}$ = 19 m. For the maximum values: $\rho_{S}$ = 10.0 Ωm, $\rho_{W}$ = 23.3 Ωm, and $t_{W}$ = 21 m. The grey shadings indicate the sediment thickness derived from the collocated SBP line.

**Figure 15 sensors-21-08053-f015:**
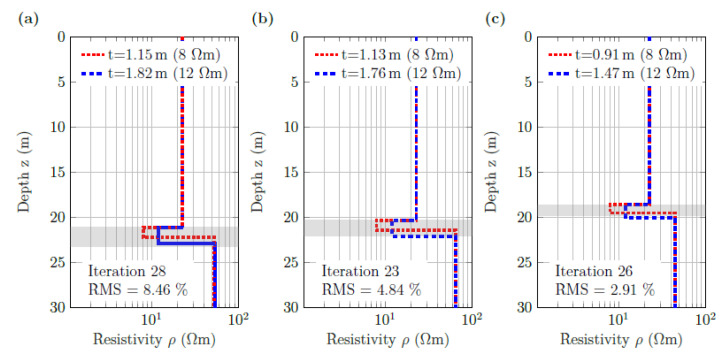
Layered resistivity models for ERT data measured at the intersections between ERT line Nah13 and SBP profiles (**a**) L3 NS, (**b**) L4 SN, and (**c**) L5 NS of section A (Figure 1b). The two inverted models correspond to two fixed values of the sediment resistivity (8 and 12 Ωm); RMS errors indicated in the respective panels are the same for both models. The grey shadings indicate the sediment layer thicknesses derived from collocated SBP profiles.

**Figure 16 sensors-21-08053-f016:**
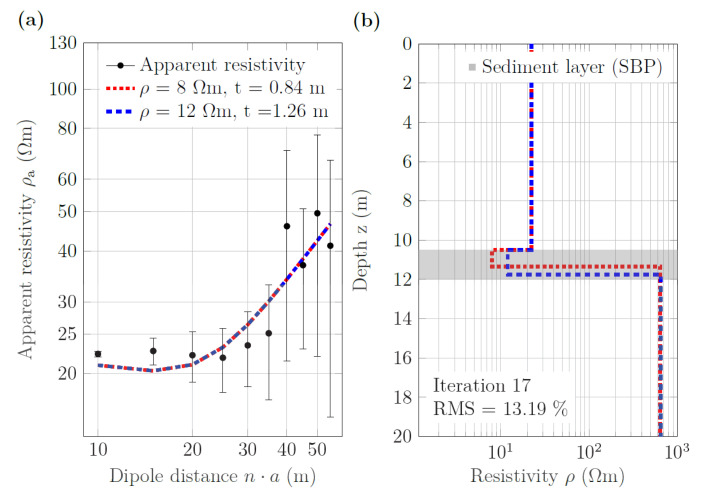
One-dimensional inversion of measured ERT data corresponding to the ten current dipoles of section C along ERT line Nah6. (**a**) Observed (black dots) and calculated apparent resistivity sounding curves. (**b**) Layered resistivity models for two different values of the fixed sediment resistivity; RMS errors were the same for both models. The grey shading indicates the sediment layer thickness derived from the collocated SBP line L4 SN.

**Figure 17 sensors-21-08053-f017:**
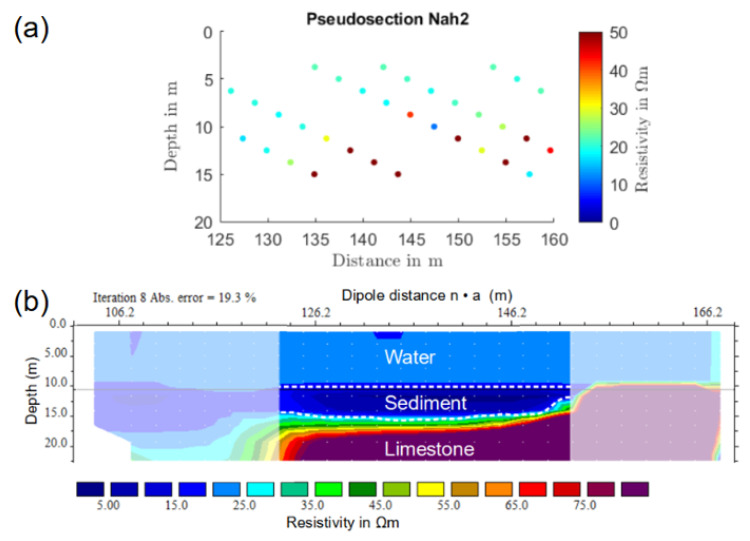
(**a**) Apparent-resistivity pseudo-section and (**b**) 2D resistivity model corresponding to location C along ERT line Nah6. Parts of the section, for which the data coverage was low, are blanked. In the central part, which had a good data coverage, a sediment layer of 5 m thickness was visible.

## Data Availability

All raw and processed data of this study (and some additional data not discussed here) are available at Zenodo (will be published upon acceptance) along with the MATLAB scripts used for the 1D inversion and to prepare the visualizations presented in this article.

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
