# Peer review of "Evaluation of Lake Sediment Thickness from Water-Borne Electrical Resistivity Tomography Data"

_sensors, 2021, doi:10.3390/s21238053_

Round 1

Reviewer 1 Report

In this work, the authors present an interesting method to characterize sediment layers on a lake in southern Mexico by using water-borne electrical resistivity tomography measurements. In order to improve the estimate of the sediment thickness at the lake bottom, they apply constraints on water depth and water resistivity values obtained from echo sounder. Furthermore, the sediment resistivity obtained from laboratory measurements on sediment samples is also used to constraint the inversion process.

The study is completed with a modelling study to check the ability of the ERT measurements to detect sediment layers at different water depths.

In my opinion, the manuscript fits well with the journal aims and could be of interest for the readers. However, bearing in mind that many more ERT profiles have been carried out in the area, the authors might consider showing more results, even a 3D model of the sediment layer.

Reviewer 2 Report

The manuscript entitled “Evaluation of Lake Sediment Thickness from Water-Borne Electrical Resistivity Tomography Data” implements water-borne Electrical Resistivity tomography measurements on lakes as a complementary technique to standard exploration methods.

Dear authors, I would like to congratulate you on your manuscript. I find it well prepared, structured and presented. I have no further comments on your manuscript.

Reviewer 3 Report

The manuscript entitled " Evaluation of Lake Sediment Thickness from Water-Borne Electrical Resistivity Tomography Data" presents a complementary method based on water-borne electrical resistivity tomography (ERT) measurements on a lake in southern Mexico. The authors present adequate results and discussion, however, it is necessary to add information in the Materials and Methods section, as well as in the results and discussion (aspects are indicated in the attached document). There is a bibliographic citation that is not in the References and more information is needed in Figure 7.

Reviewer 4 Report

The complementary method based on water-borne electrical resistivity tomography (ERT) measurements was proposed in this paper, which can provide a comprehensive image of lake-bottom sediments. However, this manuscript has several major issues such as inadequate description on principle, preventing it from being considering for publication at its current form.

Generally:

  1. The reflection seismic methods are often used in the detection of sediment thicknesses, a method combining shallow seismic exploration and ERT was proposed to detect the thickness of sediment in this paper. The further explain about the combination is necessary, such as the results of reflection seismic methods were used as a priori information for ERT.
  2. Reflection seismic methods, electrical measurement and laboratory experiments were carried out, and the relationship between them should be represented by a flowchart.
  3. The survey covered the entire area of Lake Nahá, and the reasons for selecting three eastern sections A, B, and C (Figure 1b) need to be supplemented. Moerover, the results of section C are listed without the discussion of sections A and B in Fig. 16.
  4. The principle and data processing of reflection seismic methods should to be further supplemented, such as Fig. 7. How to consider the influence of interference factors on waveform? How to consider the error caused by the small sediment thicknesses?
  5. The correlation between the electrode spacing and the depth of the test point should be further explained, and whether the larger electrode spacing such as 5 m has an impact on the test of sediment thicknesses.

Minor comments:

  1. The name of each red area should be indicated in detail in Fig. 7.
  2. Line 301: “Figure 8a” or “Figure 10a” should be corrected.
  3. The color bar in Fig. 2(a) is missing.
  4. It would be better to add the electric field distribution and test process in Fig. 4.
  5. The symbols of depth value in figures should be unified, such as Fig. 15 and Fig. 16.

Round 2

Reviewer 4 Report

I thank the authors for critically addressing the comments. The manuscript has improved significantly from the original submission. And it may be accepted subjected to approval from rest of the reviewers, editor and a thorough proofreading. However, the following minor revisions and questions should be made or explained by the authors.

Minor comments:

  1. The start point, end point and coordinate axis of the survey line are not marked in Fig. 1.
  2. The method of dealing with noisy data collected by SBP and ERT should be added.
